# Strategic Directions: Evaluation of Village Development Strategies in the Case of Applicants for the Hungarian Village Renewal Award

**Szabolcs Bérczi \*, Zita Szabó and Ágnes Sallay**





Doctoral School of Landscape Architecture and Landscape Ecology, Hungarian University of Agriculture and Life Sciences, 2100 Gödöllő, Hungary; szabo.zita28@gmail.com (Z.S.); sallay.agnes@uni-mate.hu (Á.S.)
\* Correspondence: berczisz@vzm.hu

**Abstract:** Village roles have changed significantly in Central Europe over the last century and a half. In our article, we mainly deal with the conditions in Hungary. Based on the relevant literature, we follow the changing role, problems and presence of Hungarian villages. Our research focuses on evaluating village development strategies; thus, an essential part of the article is the presentation of the European and Hungarian village renewal movement, as the 50 settlements examined are also part of the settlements launched at the Hungarian Village Renewal Award competition. In this research, the 50 settlements were divided into three groups according to their role in the settlement network. The settlement group analysed their development priorities by summarizing the Hungarian Village Renewal Award applications. As a result, it was found that the development directions of the villages belonging to the individual settlement groups can be well separated from each other. The choice of the settlement development strategy is greatly influenced by the distance from the central settlements and the settlement network situation. We compared our results with the analysis of the strategies of some foreign villages (located in the former socialist bloc) and then examined the Hungarian village surveys of the last century and a half, focusing on land use changes and their role in development. As a result of the analysis, it became clear that the importance of land use in the life of villages in the initial period decreased spectacularly over time and was replaced by employment and the role of the settlement network. The main result of our research is that we have proven that the strategic priorities of village development can be grouped based on the position of the villages in the settlement network, and the priorities are mainly determined by the size of the central settlement and the distance from it.

**Keywords:** village; strategy; development priorities; land use

## 1. Introduction

In the last century, the situation and role of villages have undergone a significant transformation, both in the western part of Europe and in the former post-socialist area. Villages have taken on a new role, looking for the correct answers to the challenges of the 21st century. Accordingly, a new type of development measure is needed among small settlements.

In our article, we aim to analyse the villages' strategic development priorities and present and interpret the renewal and development plans of the increasingly tricky small settlements and their specific development elements. Knowledge of these elements can help both professionals and decision makers to develop proposals for development programs at higher territorial levels. In our article, we are now looking for the answer to the question of which development elements and priorities determine the strategy of each village and to what extent the situation of their settlement network influences this. This article focuses on the analysis of Hungarian villages. As a result, we draw general conclusions from their results.

To gain an accurate understanding of the situation, we considered it necessary to present the history of the villages to the present day, including the changed circumstances, which may explain some development decisions, even seemingly irrational. Our studies also paid particular attention to the resources on which the villages base their development. To this end, we looked at the factors based on how the surveys of the last 150 years grouped the villages and which development factors were emphasized for each type of village. We did this because, in our view, arable land as a local resource can be an essential element in the strategic development of small settlements.

However, our primary goal is to analyse the village development strategies by examining the applications of the villages that participated in the Hungarian Village Renewal Award. Therefore, it is essential to present the details of the Hungarian (and European) village renewal movement and the related Village Renewal Award. Without this knowledge, the reader may be confused by the grouping system of our results.

The structure of the article and the logical connection of each Section are shown in Figure 1.

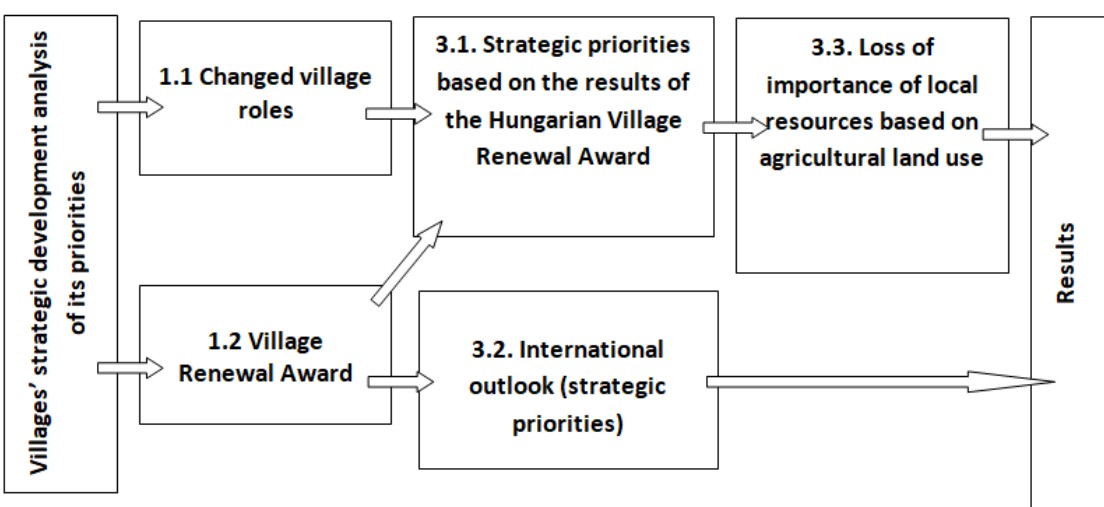

**Figure 1.** The logical connection of the researched topics.

*1.1. Changed Village Roles*

As a result of socio-economic changes in the last century, the settlement network in Central Europe has changed significantly; thus, the traditional role of villages has also changed. The villages of the countries of the socialist bloc underwent changes that changed from country to country after the change of regime. However, the geographical location of the settlement (deprived area or urbanisation) and the land ownership structure played a decisive role in the changes. Development opportunities are determined by political heritage, but also by the weakening of the role of agriculture in the economy [1], urbanisation [2,3], the backwardness or rapid development of certain areas [4], and sustainability in agriculture [5–7], which define research directions. In addition, the European Union's agricultural [8,9] and rural development [10] directives, which already include complex development aspects [11], set out development opportunities. Research in different fields of science also shows that supporting different local initiatives can significantly contribute to rural development [12,13].

Despite the similar political system, different trends can be identified in each country during the changes in the settlement network [14]; thus, we briefly present the changes in the settlement structure in Hungary since the 19th century, as this is essential for analysing and understanding the current situation. Furthermore, we want to present the changed situation through the processes in Hungary, where given that Hungary was located on the eastern side of the Iron Curtain, the process created more extreme situations than the European average.

Hungary was traditionally characterized by agricultural production in the 19th century. A system of large feudal estates characterized the century, and this estate structure was and survived until the middle of the 19th century.

Before the First World War, the vast majority of the Hungarian rural population lived off agriculture [15], under challenging conditions, two of which were due to the underdevelopment of the large estate system and industry. This situation did not improve between the two world wars [16]. According to contemporary interpretations, the village was the scene of traditional peasant society and agricultural production. However, it was gradually transformed into a new, more diverse form of settlement [17]. However, the "village = agricultural settlement" formula, due to socialist economic and settlement policies, became invalid even before the change of regime [18,19]. Rural society was no longer composed exclusively of the agricultural population [20], and after the change of regime, the process intensified further.

Due to the communist takeover after the Second World War, the self-determination of local governments ceased to exist, several settlements came under the joint control of the council, and collectivizing agricultural policy and intense industrialisation exacerbated territorial differences [21]. The general phenomena of the first stage of socialism are industrialisation at a stormy pace, the reorganization of agriculture, the liquidation of homesteads (and the creation of new homesteads), the stratification of employment, and the rapid, accelerating growth of cities. As a result, the change in the state of the villages became the most characteristic. There were no mature types of transforming village at that time. Their general feature was only the separation of the traditional unity of settlement, agriculture and peasantry [22].

In socialism, the most crucial sector of the economy was industry, and in 1970, 10–15% of the village population was industrial and one-third mixed [18]. The transformation of villages was also significantly influenced by the effects of settlement and social policy: the construction of the socialist economy (collectivisation, industrialisation) favoured certain areas, and according to the principles of settlement network planning, the differences in supply increased; thus, the less-favoured areas due to the structural network and the poor condition of the housing stock are becoming increasingly disconnected [19]. By the end of socialism, however, due to the unified treatment of socialism, the villages were increasingly losing their former character [23].

Following the change of regime, the 1990 Local Government Act gave settlements an entirely new legal status, the municipalities were "freed" from the subordination of the territorial level, the principle of self-government considered the possibility of self-government as a fundamental part of the democratic system [24,25], and municipalities, regardless of size, were granted complete municipal independence.

As a result of self-sufficiency, local governments have made significant economic and infrastructure development progress. However, the value of improvements is diminished because most municipalities have only achieved partial results without a comprehensive renewal strategy and have stalled since initial improvements [26]. The great euphoria of independence was thus soon followed by rapid sobriety [27]. However, in the first half of the 1990s, there were signs of insolvency in some elements of the fragmented structure, especially in small villages with structural problems [25].

The framework of the development of the settlement network has changed radically. The separation of individual villages and the formation of new villages have been facilitated [28], the population of tiny villages has become more fragmented, the number of dwarf villages has increased [29,30], the number of local governments has increased to over 3000 and the competition of settlements has intensified. "Villages have entered the free market of their settlements", in which their relative position is determined by their geographical location, their endowments and the local policies that exploit them [19,31]. Thus, the introduction of free local government and normative financing fundamentally changed the previous structure. As the local government was associated with rights and

obligations, there was a great deal of tension between local governments between the elements of a fragmented system and the allocation of resources [25].

Following the regime change, the social processes caused by the sudden freedom also rearranged the roles in the settlement network [32]. The development and population-absorbing power of the cities had severe consequences for the villages outside the agglomerations of the big cities: the small settlements had to face the worsening processes of emigration and ageing: primary care was lost as well as services for the local society, society was ageing and services had deteriorated further [20,33]. Patrick Drudy called this process the "cumulative cycle process" [34]. Thus, the prevention of unfavourable demographic processes and the strengthening of local society became the primary goals of the survival and development of villages.

The most characteristic features of the transformation of the settlement network were geographical deconcentration and territorial differentiation, which resulted in regional transformations. Several rural areas rose in parallel with the classical suburbanisation processes [35,36].

As a result of the settlement network and the urbanisation of society, the emigration of young people to cities has intensified. The population of rural villages has decreased, and society has become older [37]. However, in addition to the changing role of villages, the social demands placed on them have also changed. The "urbanisation" of the villages, the supply expansion and the improvement of the quality of locally available services became basic expectations. At the same time, in the age of digitalisation, the village can also be seen as the opposite of the accelerated urban way of life: the calm living environment and the need for a "rural" way of life are becoming more and more critical [38]. The above two opposite processes also set the villages on a new development path. Due to the changing roles and expectations, the new development directions and strategies could be the breaking points of the villages, through which the individual small settlements could become successful.

Villages need to place more and more emphasis on their development as a result of urban competition such that they can be an attractive alternative to the city for the population. To this end, (conscious) settlement development has become essential for the villages. Their renewal and development can be seen as a potential living space by the locals and the people wishing to settle here. The decline and then renewal of villages (and rural areas), the conscious development of settlements, and, over time, the growing number of leading-edge villages have given rise to a village renewal movement that maintains villages (and rural areas). Aimed to preserve them, increase their vitality, and develop them sustainably [39]. Along with these principles, the Hungarian and European Village Renewal Awards are organized every two years, described in detail in Section 1.2.

*1.2. Village Renewal Award*

From the end of the 20th century, the renewal of villages and rural areas, village and rural development has become more and more critical [40], the principles of which were enshrined in the 1996 Cork Declaration [41], and which were further developed in 2016 [42]. The implementation of the Cork Declaration is being developed by the European Union's Directorate—General for Agriculture and Rural Development within the framework of the Common Agricultural Policy [43]. Today, village renewal has also become a domestic and European movement. As a result, dozens of villages can be set as an example in front of other small settlements. In our research, we deal with these villages: we want to present the development of those villages which, recognizing their changing roles, were able to provide an appropriate response to the challenges of the new millennium and serve as a positive example for other settlements.

Our research examined the results of the settlements that participated in the Hungarian Village Renewal Award competition. The Hungarian Village Renewal Award competition grew out of the village renewal movement launched by the European-based European Rural Development and Village Renewal Working Community (Europäische ARGE Lan-

dentwicklung und Dorferneuerung) [44,45]. The European Economic and Social Committee has also prioritised this issue, as exemplary initiatives, good practice and the dissemination of good practice are essential for the regeneration of Europe's rural areas [46], and the Village Renewal Movement seeks to motivate villages in [47].

The European Community organizes the European Village Renewal Award for Rural Development and Village Renewal. (The Working Community operates voluntarily as an international NGO, independent of EU institutions and bodies.) The European Village Renewal Award has been announced since 1990, with an international panel of experts giving an opinion every two years on the success of each applicant's work [39].

The application initially focused on two strategic themes: the situation of villages (landscape, settlement, cultural heritage and infrastructural characteristics) and related developments (cultural landscape, agriculture, building stock, renewable energy, local quality of life, or social and cultural institutions). In 1998, the call for proposals was modified. First, the development goals were divided into ten, later seven and eight, and since 2012, nine topics. These were the most important strategic areas of village development, and applicants had to present their development results in these areas [48].

In addition to presenting thematic developments, comprehensive conceptual planning and strategic processes, sustainable development and partnership have become increasingly important in the call for proposals, focusing on individual initiatives, dialogue between politicians, experts, public authorities, local people and regional cooperation.

Each European Village Renewal Award competition has its motto, and compliance with it is a priority during the competition. The competition's motto has constantly been changing over the last 30 years. It is clear from each title that the challenges the advertisers were looking for have definitive answers in the current period (Appendix A). (Of particular interest is the 2020 motto that local responses to global challenges are the biggest challenge).

The villages participating in the European Village Renewal Award are evaluated according to a complex system of criteria, taking into account the initial situation, development goals and processes, and the individual development projects in each thematic area.

The Hungarian competition is also closely related to the international village renewal competition. In general, we can discuss the embedding of the Hungarian competition in the European Village Renewal Award competition, considering that after the Hungarian competition, the winner will represent Hungary in the European competition. The Hungarian Village Renewal Award competition is announced every two years. The applicant settlements document the life course and development goals of the villages for the development period, as well as the programs and measures taken for these purposes. The most important aspect is how complex the programs implemented during the development serve the development of the village, and whether they provide an appropriate answer to the challenges and problems raised by (the village). The thematic programs examined during the development are the following:

- Strengthening environmentally friendly agriculture and forestry;
- Responsible and environmentally friendly resource management, use of renewable raw materials;
- Maintaining local supply and employment opportunities and creating new ones;
- Renovation of the old building stock worth preserving, construction of new, high-quality buildings;
- Modern social institutions, creation of opportunities for socio-cultural life;
- Strengthening the identity and self-awareness of the local population;
- Developing the skills and motivations of the population to develop their commitment to the community;
- Promoting economic, social and cultural equality for all ages, nationalities and minorities;
- Networks, inter-municipal relations.

The grouping of the programs into thematic areas as required above requires a tight strategic approach when compiling the application documents (and during the decision-making process). In addition, it makes it easier to compare individual villages.

## 2. Materials and Methods

In the course of our research, we worked with the data of 50 villages launched in the Hungarian Village Renewal Award competition (Figure 2). The 20–40-page application materials of each village contain statistical data according to the Hungarian Central Statistical Office (KSH), local characteristics (e.g., several local associations, NGOs and number of members) as well as development goals and implemented improvements. The application materials show the aggregation and systematization of data that also contain the national register (KSH) data; thus, their separate collection [49] has not become necessary.

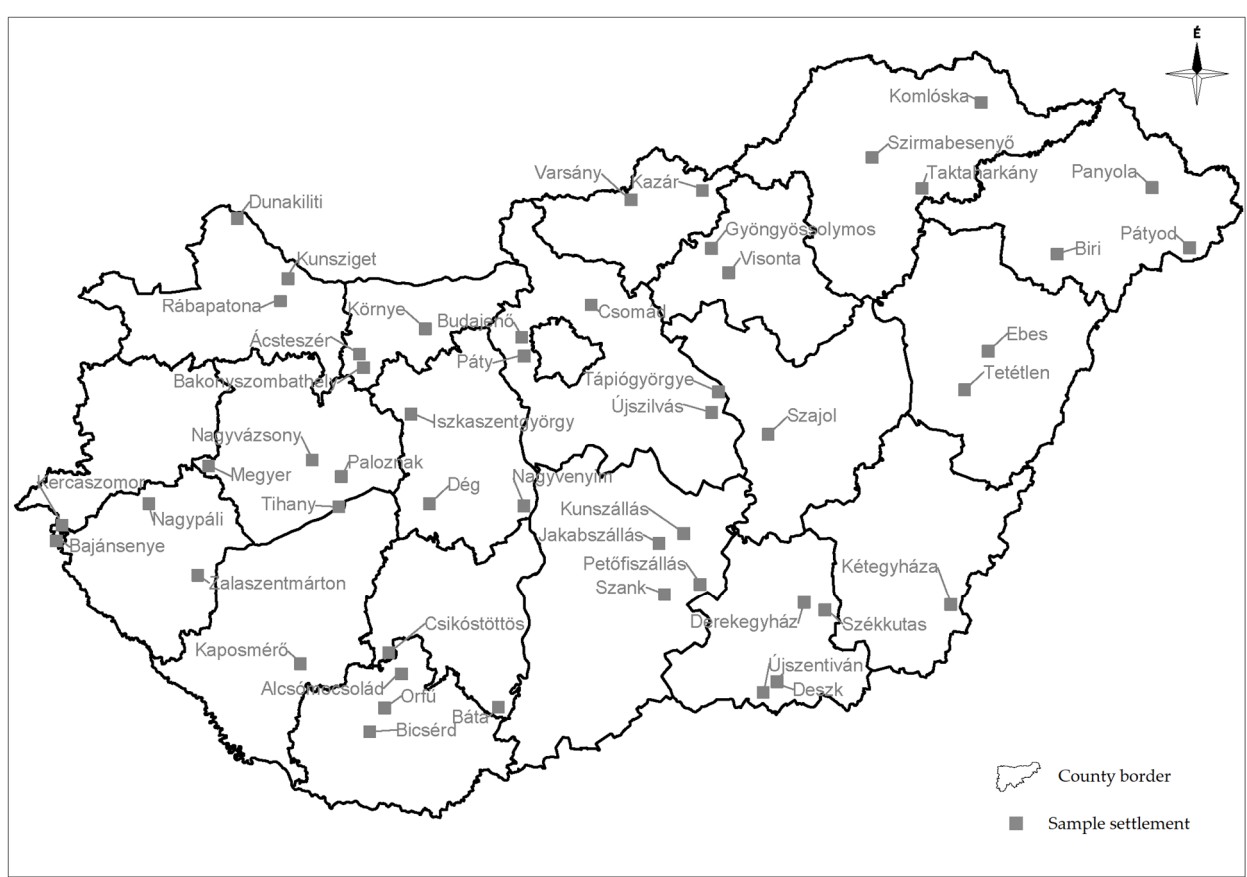

**Figure 2.** Location of sample settlements.

According to the literature data, the 50 villages represented a diverse set of settlements that could represent the Hungarian village types. Therefore, for the analysis, the sample settlements needed to have different villages in terms of geographical location, settlement size and the settlement network's role.

One of the hypotheses of our research is that the nature of the development strategy, the selection of the development directions and the essential characteristics of the strategy are determined by the regional location of the villages and their role in the settlement network. Accordingly, we classified the sample settlements into three categories: settlements located in metropolitan agglomerations (in the vicinity of county capitals), settlements located in small-town catchment areas, and settlements located in depopulated areas; the latter category included all the settlements in the catchment area of small towns with less than 10,000 inhabitants. Grouping was performed according to the central settlement of each settlement group, differentiated according to the function (and size) of the central settlement. Thus, the basis of the grouping was not the villages but the cities that represent the spatial organizing power of the villages. (The grouping principle was also supported by the results of the settlement network research carried out by VÁTI (Urban Planning Office, later VÁTI Hungarian Regional Development and Urban Planning Nonprofit Ltd., Budapest,

Hungary) in the 2000s, according to which small towns and new settlements in terms of their function are in many cases unable to perform the task of a regional organization. Thus, the availability of actual centres from smaller settlements in the vicinity of these "appearance" cities becomes problematic, and the quality of life in villages located in functionally deprived areas decreases [31].

In our research, the classification of settlements into three groups was validated using KSH basic data (KSH serial number, "success index" and pattern analysis of basic statistics of sample villages and their neighbours-mean, standard deviation, median, minimum, maximum, confidence interval) [50]. Thus, the individual examinations were performed separately for the groups. After the classification into three groups, 17 of the examined settlements belong to metropolitan agglomerations, 20 to small-town catchment areas, while 13 are located in areas without urban areas (Figure 3).

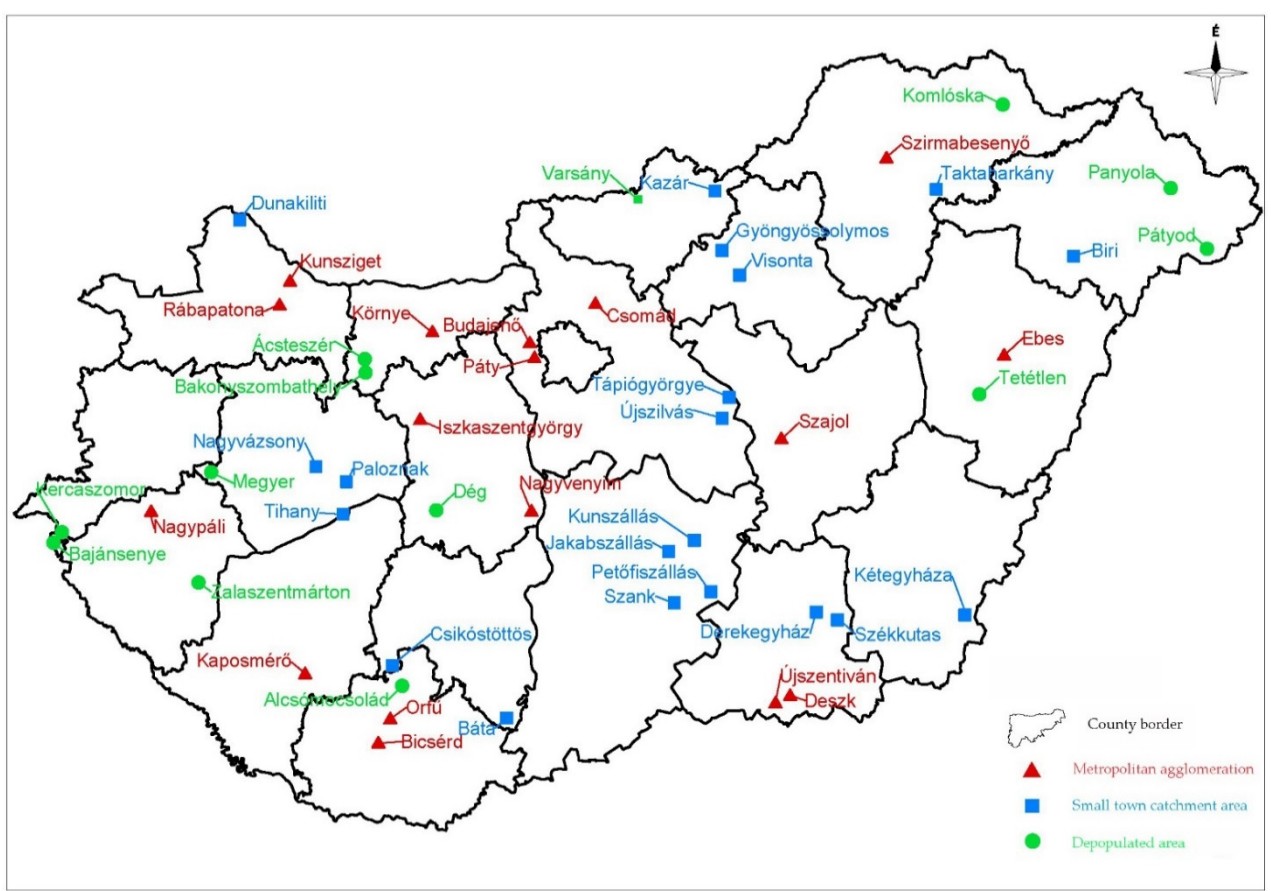

**Figure 3.** Location of sample settlements by groups of settlements.

An overview of the 50 entries in the Hungarian Village Renewal Award competition was necessary to analyse development strategies. (During the analysis, the results of the settlement groups were evaluated in general and separately according to the evaluation criteria.) The analysis of the tender participants in 2005–2019 used its tender documents and evaluation results. Based on these, the intervention's justification, effectiveness and quality had to be assessed for all thematic areas; they could be rated as poor, average or exemplary. The evaluation was performed with a value on a scale of 0–10 points per subject area. (The panel of experts has been scoring the applications since the 2017 competition, before which the grading was based on an oral evaluation by the committee members. Applications before 2017 in the absence of a score were evaluated based on written application materials and personal experience).

According to the critiques, the interventions of each village by thematic area were considered successful if they were exemplary in all respects—justification, effectiveness,



quality—and the development strategy was successful if exemplary interventions were implemented in all thematic areas.

Following the analysis of village renewal strategies, we analysed the village surveys in Hungary (at the national level), with a particular focus on land use. In the analysis, the methods of five nationwide surveys, which faithfully reflect the settlement population of the given period, were listed, emphasizing the methodological changes of the individual surveys and the modification of the cluster-forming variables of the settlement groups defined by the surveys.

## 3. Results

### 3.1. Strategic Priorities Based on the Results of the Hungarian Village Renewal Award

Our research focused on which areas the villages had hoped to develop in recent decades, which areas they had focused on, and what thematic programs they had set to develop the village.

The detailed analysis covered the applications of the 50 settlements that participated in the Hungarian Village Renewal Award competition, which were examined primarily based on their role in the settlement network and the emphasis of their development strategy.

The exact names of the thematic programs are presented in Section 1.2, and for the sake of simplicity and transparency, these programs will be referred to as follows:

- Agriculture and forestry;
- Sustainable resource use;
- Employment;
- Quality building stock;
- Socio-cultural life;
- Local identity;
- Community building;
- Equal opportunities;
- Network connections.

Examining the applications of the 50 villages, it became clear that the settlements carried out improvements following several priorities. However, the question arose as to whether there is a common feature between the different development programs and strategies or if the presented villages are independent examples of successful developments.

As the characteristics of the development strategies and the emphasis on the individual thematic areas within the strategy are greatly influenced by the role of the village in the settlement network, the results of the villages were examined into three settlement groups presented in Section 2.

The analysis of the development strategies of the settlements for the nine thematic areas of the Village Renewal Award competition presented above confirmed the assumption that the role played in the settlement network significantly influences the choice of the development strategy. The standard features of the strategies separated according to the settlement groups have shown that, depending on the role played in the settlement network, some topic areas are given more emphasis, and others become completely insignificant and unjustified.

The results of the 50 sample settlements and the emphasis programs of the development strategies are shown in Figure 4. The percentage values according to the vertical axis of the graph show the proportion of successful settlements in terms of the topic area within their settlement category. The horizontal axis shows the application topics.

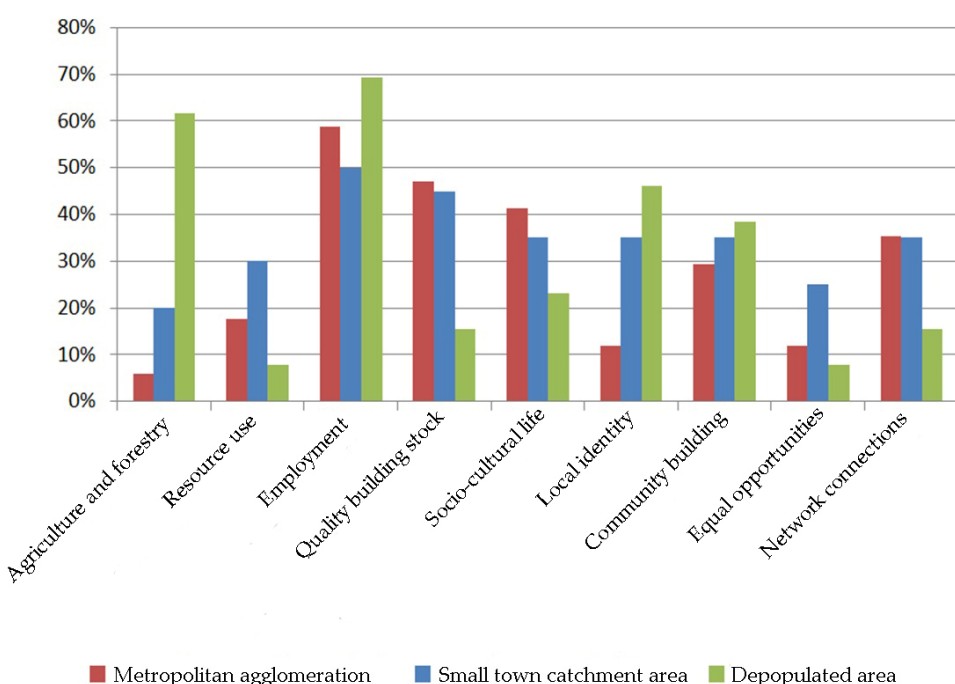

**Figure 4.** Development emphases depending on the role of the settlement network.

Based on the results, our main findings on thriving areas of village development are as follows:

- The need to develop employment is critical for all settlement groups. Employment is most important and productive in depopulated areas and agglomerations of large cities than in small towns.
- In addition to expanding employment, the villages of depopulated areas are the most successful in developing agriculture and forestry, while the settlements of the metropolitan agglomeration and small-town catchment areas are dominated by the quality development of the building stock.
- Community building and strengthening local identity is more successful in villages in depopulated areas (and small-town catchment areas) than in settlements around large cities.
- In the case of network connections, the lack of this is most pronounced in deprived areas.

*3.2. International Outlook*

Reviewing the applications in Hungary, the question may arise as to how the villages operate in the rest of Europe. In the case of European competitors, the analysis based on the settlement groups used in Hungarian villages is not necessarily expedient, as the administrative system of the settlement network in Hungary is completely different from the legal regulation of the Austrian, German, Swiss or Polish settlement system (administrative structure). In addition to the legal framework, the social differences of the last century and the natural geographical factors are also striking. At the same time, it is worth examining the results of some European competitors, because although we cannot obtain a comprehensive picture, we can gain insight into the general processes of wood-tree development.

To draw a parallel between the Hungarian applicants and the villages participating in the last European Village Renewal Award [48] we selected five German and Polish villages. We reviewed their applications: Dobkow (Lower Silesia, Poland) [51], Giersleben (Saxony-Anhalt, Germany) [52], Kadlub (Opole, Poland) [53], Rammenau (Saxony, Germany) [54] and Steinbach (Thuringia, Germany) [55]. A common feature of the villages

was that they were located in the socialist bloc before the regime change. (Thus, in essence, their starting position is somewhat similar to that of Hungarian villages.)

Among the villages, according to a similar starting point presented above, further commonalities can be discovered:

- Before the start of village renewal, the villages all struggled with the difficulties of economic and social restructuring caused by the change of regime (closure of large farms, changes in agricultural ownership, resulting in unemployment, emigration, cessation of local services);
- Each of the villages belong to the catchment area of a medium or large city (Dobkow-Jelenia Gora; Kadlub-Opole; Giersleben-Aschersleben, Strasbourg, Bernburg; Rammenau-Bautzen, Dresden; Steinbach-Eisenach, Gotha), which has a significant extraction effect;
- Except for Giersleben, the villages both belong to the administrative area of a small town; thus, they do not have their budgets. This situation has accelerated the implementation of local developments and the strengthening of local identity;
- The starting point for the development was a consciously prepared, detailed development strategy.

Examining the development processes, the unique characteristics of each village can also be found, in addition to the standard features.

Located in Sudetenland, Dobkow (Poland) has based its development on local resources, highlighting all the elements of the project to present natural heritage: environmental awareness and harmony with nature in all areas of life (volcanic study trail, lectures emphasizing the importance of biodiversity, ecotourism developments, etc.). The other development priorities preserve built and cultural values: the renovation and recycling of typical buildings in Sudetenland (restaurant, accommodation, museum and education centre) and the preservation and passing on of traditions (beekeepers, potters, camps and meetings of traditional artists). These goals can be framed by a program called the "open-air eco-museum," in which local values can be visited through a village tour [51].

Kadlub's (Poland) primary strategic goal is to improve local care and expand local services through job creation and institutional development and a conscious community building that strengthens local identity. To achieve this, emphasis was placed on measures to set up businesses and create jobs; expand and improve the quality of services provided by educational, health and social institutions; expand the range of leisure activities; intensify intergenerational programs; and create new community opportunities and spaces (sports halls, cultural and leisure centres, outdoor sports centres and event space) [53].

Similar goals have been set in Giersleben (Germany), from which any major city is easily accessible due to excellent road and cable car transport. The main goals in the village are to increase the local standard of living (high-speed fibre-optic internet, quality social care for all ages, expand leisure activities), to ensure energy independence (wind farm), and to "save" the school as a community organizing force and development, strengthening the local civil society and strengthening gentle tourism [52].

Rammenau (Germany) has traditionally been an agricultural settlement, but this has only been a partial part of the main elements of the development strategy, because agriculture here (also) provides a livelihood for only a few people. (At the same time, agriculture also played an essential role in energy production through the built-in biogas plant.) The main development goals of Rammenau are job creation (an economic area for small businesses has been created on the outskirts of the village), strengthened tourism (fishpond, baroque castle, renovation of the built heritage—blacksmith shop, prison—utilization), creation of community spaces, community buildings and active civil life (civil house, village house event space, community space in the old smithy, many non-governmental organizations, regular events, etc.) as well as continuous education of environmental awareness from childhood and the use of renewable energy sources [54].

Steinbach (Germany), located in the mountains of Thuringia, has traditionally been an industrial village (there was a knife factory employing 1000 people in the GDR); thus, they

had to face even more severe social and employment problems after the change of regime: 90% unemployment, 25% migration of the population, cessation of local shops and services. Therefore, the village has set up a joint development program with the neighbouring Bad Liebenstein, mainly to expand local employment (support small businesses: knife manufacture, brewery) and preserve tradition (cult of Martin Luther, a knife as a local symbol). As a high-quality locale, it focuses on the provision of health and social care[1], the development of community life (creation of community spaces, events), and the expansion of tourism (baroque castle, Europe-famous car race, Martin Luther cult, Steinbach knife) [55].

Reviewing the international examples, it can be stated that there are commonalities with the strategic development priorities of the Hungarian villages. However, the differences are also apparent, and the three groups of settlements defined in the Hungarian villages do not completely stand out in international cases.

As we showed in the introductory part of the paper, all foreign villages belong to the catchment area of a medium or large city, but when evaluating the strategies—compared to grouping the results of Hungarian village development strategies—some of them carry strategic elements of depopulated areas (e.g., Dobkow, Steinbach). Moreover, in the case of other settlements, the strategic elements of small-town catchment areas dominate (e.g., local identity, community building, resource use). The local identity and local community are more important in the villages located in the metropolitan areas than in the case of the Hungarian agglomeration villages. This may be due to:

- The different social customs and administrative situations in each country having created fundamentally different socio-economic situations, including different development priorities;
- Villages are mostly part of a small town (administration), and they do not have an independent decision-making body or an independent budget.

As a result of the above, the development based on independence and local society is a more vital driving force in all settlement groups than in Hungary. Thus, overall, these strategic priorities cannot be fully identified with the emphatic strategic elements of the Hungarian settlement groups.

### 3.3. Loss of Importance of Local Resources Based on Agricultural Land Use

The analysis of village renewal strategies in Hungary showed that (successful) developments based on agriculture and forestry as internal resources are present in only a tiny proportion of villages, with only a (smaller) share of local natural resources and land use prevailing, which determines the development priorities and strategic elements of villages. This is why we have reviewed the Hungarian village surveys of the last century and a half, focusing on the role of agriculture and forestry and the changes in the land use of villages.

During the land use surveys of the villages, it is worth reviewing the research that includes village surveys and village typifications. In Hungary, after the new millennium, these studies have mainly focused on the success of rural areas or the development of small village areas [33,37,56–58]. In addition to these, however, there are a large number of villages that either concentrated on a segment of Hungarian settlements [37,58–60] or conducted a nationwide survey [21,58,59]. It is interesting that during the typification of the villages, initially, the "external features" were decisive (morphology, size, population, form of farming), while later, they were due to the changes in the settlement network and the society, more complex indicators (living standards, nature of employment, population movement, development). Finally, village types were determined using complex statistical methods (cluster analysis, factor analysis).

In the present study, the villages were grouped in the manner defined in Section 2. Therefore, during the examination of the changes in land use, we are interested not in the changes in the village types but in the change in the classification methodology and the shifts in the development factors.

We have used five nationwide surveys of village research over the past 150 years, which are well differentiated over time. (The results of the first and second surveys were

processed together.) Therefore, by ranking the results of the surveys, it is possible to show with well-defined characteristics how the aspects of the classification changed during each survey.

The first two surveys examined dates back to the time of the Reformation, the first was made by András Vályi[2] at the turns of the 18th and the 19th centuries [61], and the second Elek Fényes[3] in the middle of the 19th century [62]. Both works present the individual villages at the settlement level, giving a detailed description of the contemporary land uses, the quality of the land, and the crops grown.

In the 19th century, the natural endowments determined the possibilities of farming and thus of the given settlement. (In addition to the social and agricultural characteristics, the description of the villages in the examined villages highlights the location of only one castle, castle, inn or particular function.) Based on the works by András Vályi and Elek Fényes, the villages are considered as forest, meadow, vineyard, and livestock, with the simultaneous indication of unique local conditions (e.g., tobacco growing, vegetable growing, beekeeping, presence of a swell, castle or castle).

It was quite a long time, more than 100 years, until the date of the following national-level survey, which also determined the territorial characteristics, after the geographical dictionary (country description) of Elek Fényes in 1851. However, before the Second World War, the development of villages was still determined by agricultural conditions; thus, we cannot discuss more severe land use changes at the settlement level. (From Ferenc Erdei in 1940 entitled *Hungarian Village*, in his book, he separated the villages according to their agricultural ownership and social forms. In this idea, social aspects also appear in the grouping of villages. However, in Erdei's work, the village is still a type of settlement related to agriculture (peasant farming), and the typical peasant village is the "type of village that is usually cited as a village" [63]).

This situation was changed by the socialist takeover, which placed agriculture on a new footing and emphasised changes in the settlement network through industrial relocation and the reorganisation of agriculture. As a result, the status of settlements has entirely changed in many cases, and the results of this process are well illustrated by the work of Pál Beluszky[4] and Tamás T. Sikos[5] [64].

The authors aimed in their book (titled *Village types in Hungary*, from 1982) to examine the villages at the national level, assess and classify the condition of the villages, and differentiate the types of villages. (Regarding the changes in local resources and energy in the villages, despite the long time that has elapsed, the data from 1851 and 1982 can be considered successive periods in our study.) Therefore, the method of cluster analysis was chosen for the classification, during which eight groups of factors were identified:

- the natural environment of the villages;
- the place of villages in the settlement structure;
- the economic role of villages;
- the development of the role of the primary care provider in the villages;
- direction and pace of settlement development;
- the traffic situation of the villages;
- the artificial environment of the villages; housing;
- the level of the general development of the villages.

Based on the cluster analysis, the distinction between settlement types is based primarily on the occupational structure and on population change and/or settlement size. As a result of the typification, the research distinguishes seven village types (25 clusters) (Figure 5, Table 1).

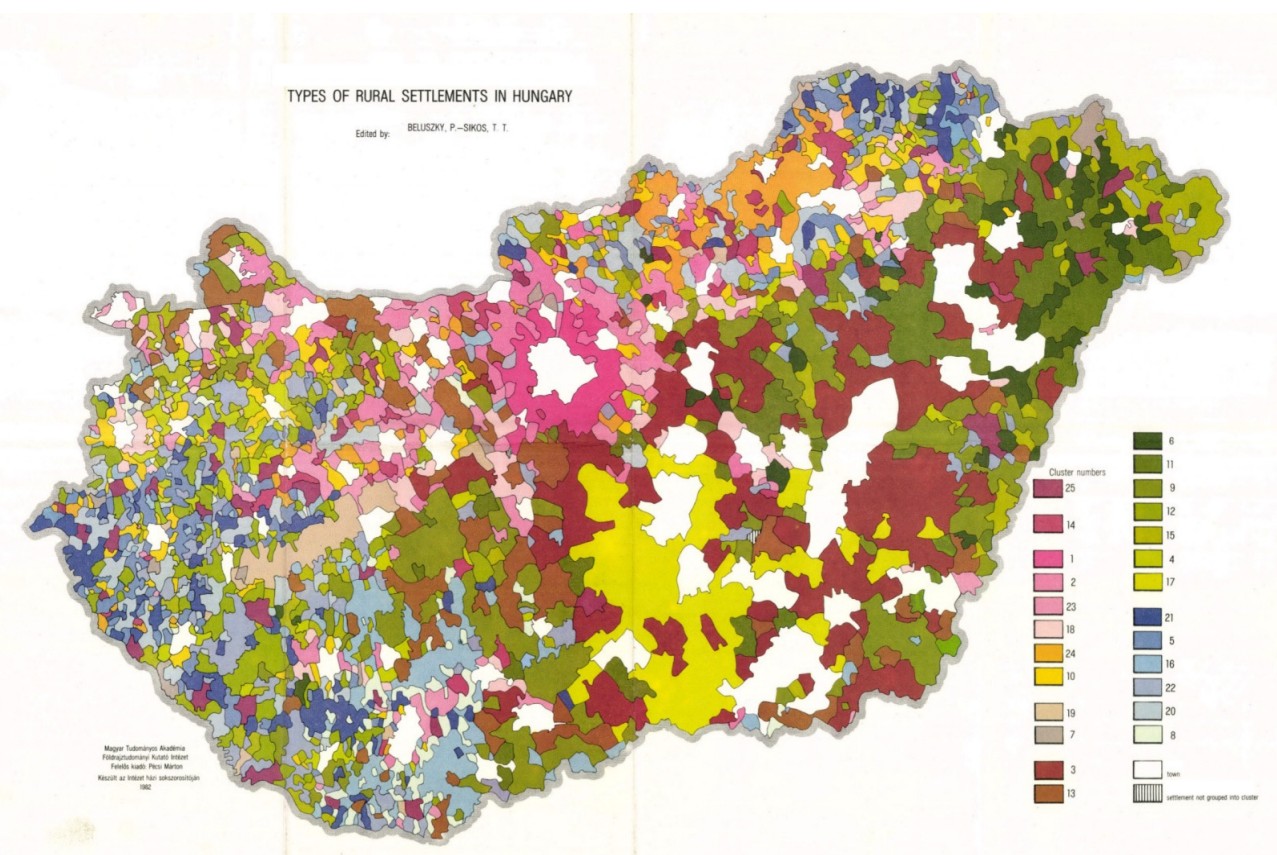

**Figure 5.** Types of rural settlements in Hungary, 1982 [64].

In the characterization of village types (subtypes), the dominant economic sector plays a significant role, as shown by the names of the individual clusters (agricultural nature, agro-mixed functions, industrial municipalities, industrial-mixed and industrial-tertiary employment). It can be seen that the focus has shifted from land use to employment.

The next stage of our study is the analysis of the same pair of authors in 2007, the main aim of which is to present the changes of villages and the reclassification of village types at the beginning of the third millennium, especially after (and as a result of) the regime change [21]. As a result, the criteria for village classification also changed, during which seven main points and 27 variables were defined, as follows (number of variables in parentheses):

- land use, natural resources (1);
- the place of villages in the settlement structure (3);
- the economic role of villages (9);
- the traffic situation in the villages (1);
- basic provision of villages (2);
- the demographic and social situation of the villages, income and wealth relations (8),
- the pace and direction of settlement development (3).

As a result of the cluster analysis, the classification of village types is based on the role played in the settlement network, the labour market situation and the population change. The authors again distinguish seven village types (and 25 clusters) (Figure 6, Table 2).

**Table 1.** The name of the types of rural settlements in Hungary, 1982 [64].

| Ordinal Number | Village Type Name | Clusters |
|:---:|:---|:---:|
| I. | Small villages with a rapidly declining population, with no primary education, with unfavourable living conditions, with one-level functions | 5, 8, 16, 20, 21, 22. |
| II. | Medium-sized villages with traditional village functions and agricultural (additionally industrial or tertiary) occupational structure | 4, 6, 9, 11, 12, 15, 17 |
| III. | Large and giant villages with mixed agriculture | 3, 13 |
| IV. | Centrally located, urban-type municipalities with an industrial-tertiary employment structure | 25 |
| V. | Population industrial villages, with a very rapid population growth, an urban environment, sometimes with an urban function | 14 |
| VI. | Rural settlements of agglomerations and residential areas | 1, 2, 10, 18, 23, 24 |
| VII. | Municipalities with special roles | 7, 19 |

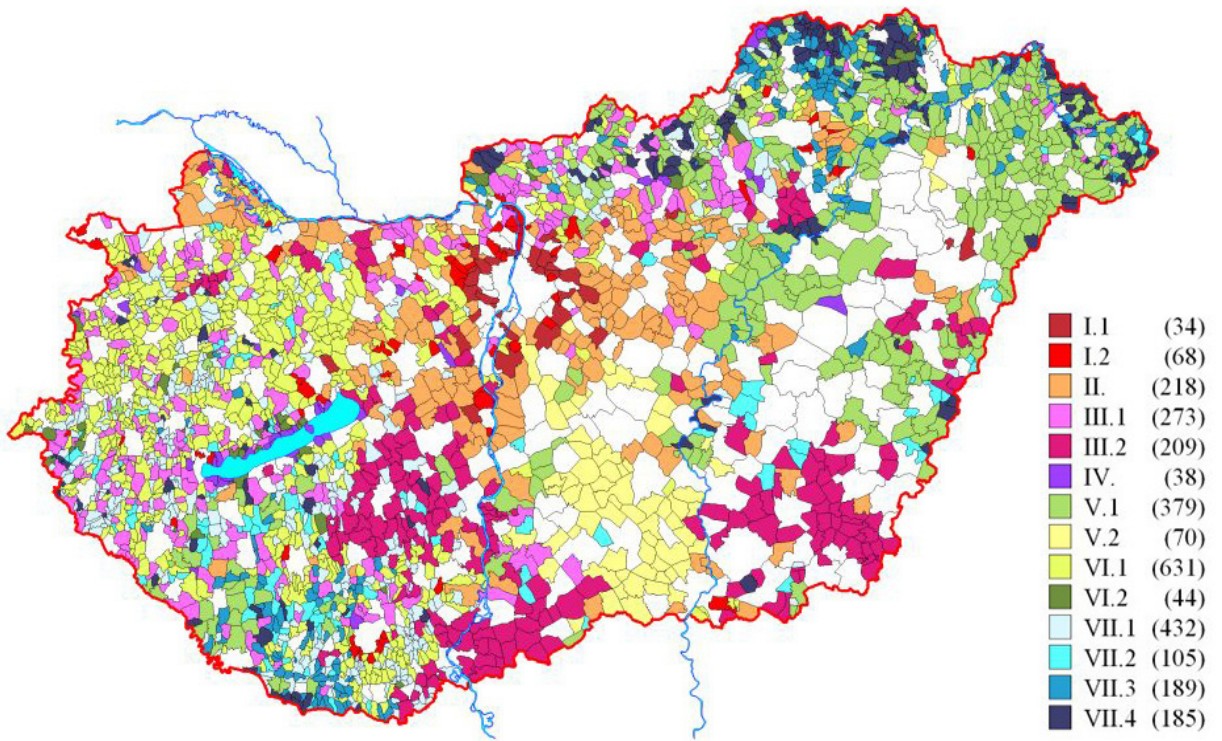

**Figure 6.** Village types in Hungary, 2007 [21].

**Table 2.** The name of the village types in Hungary, 2007 [21].

| Ordinal Number | Village Type Name | Clusters |
|:---:|:---|:---:|
| I. | Inner zone of agglomerations | 1, 12, 17, 19 |
| II. | Municipalities belonging to the outer zone of agglomerations | 4 |
| III. | Smaller, stagnant-moderately declining residential and mixed-use villages | 14, 22 |
| IV. | Villages and spas with a tourism role | 6, 7, 9, 24 |
| V. | Medium-sized villages with an unfavourable labour market situation, sometimes with a significant agricultural role or a peripheral population | 15, 16 |
| VI. | Small villages with a good labour market situation and a stable society, with a residential and tourism sector | 8, 11, 18, 20, 25 |
| VII. | Small villages with poor labour market situation, declining population, disadvantaged, distorted demographic-social structure (disadvantaged small villages) | 2, 3, 5, 10, 13, 21, 23 |

Due to the socio-economic processes that took place after the regime change and the significant transformation of the settlement network, the emphasis on the separation of new village types is no longer on land use or employment stratification, but rather on demographic processes and the labour market situation. It can be stated that while earlier agricultural (later industrial) analysis dominated the clusters, after the turn of the millennium, the strength of the role of the residential function is the most crucial difference.

The last element of our analysis of the change in local resources is the national settlement cluster commissioned by the Ministry of the Interior, which was created in 2019 following the work of Miklós Illésy, Judit T. Nagy and Róza Számadó [65]. In this analysis, the authors divided the settlements into two groups: settlements with a population of over 2000 and less, and the statistical analyses were performed separately for the settlement groups. Cluster analysis was chosen as the classification method, using eight categories and 23 variables, of which the success criteria were cluster-forming variables.

The variables fell into the following categories:

- success indicators (criteria);
- demographic variables;
- variables measuring economic strength;
- variables measuring geographical location;
- variables measuring the development of the civil sector and cultural life;
- variables measuring the development of public service infrastructure;
- variables measuring the development of municipal co-operation;
- variables measuring the online presence of municipalities.

The cluster analysis resulted in three clusters for each settlement with less than 2000 inhabitants. The study summarizes the settlements with less than 2000 inhabitants under the collective name "small settlement", of which 2372 settlements belong to the clusters, and 781 settlements belong to the clusters with more than 2000 inhabitants. The names of the clusters are as follows (in brackets the symbol of the cluster and the number of settlements included):

- lagging small settlements (2000−/1; 1353 settlement);
- hybrid dwarf villages (2000−/2; 124 settlement);
- booming small settlements (2000−/3; 895 settlement);
- disadvantaged settlements (2000+/1; 341 settlement);

- settlements in the catchment area (2000+/2; 129 settlement);
- less attractive subcentres (2000+/3; 311 settlement).

Clusters are defined by the dynamics of the development of settlements. In addition to their role in the dominant network of settlements, economic, social and cultural aspects are essential. As a result, the results of the land use characteristics are minimal (almost non-existent).

## 4. Discussion

Based on the examination of the tender results of the Hungarian Village Renewal Award, we found that a close correlation can be observed between the development emphases and the distance from the centre settlements of the settlement network.

The conscious development of community buildings and local identity plays an increasingly important role in moving away from big cities. Accordingly, the development of these areas is more successful than in the villages of depopulated areas (and small towns). However, this is not the case in the settlements of the metropolitan agglomeration because, although financial resources are available, due to the different needs of a diverse society, in most cases, we cannot speak of a classical community; thus, in many cases, in short, they focus on improving the quality of life. A similar trend is valid for the utilization of network connections. This is not strong in any of the settlement groups. However, the majority of network connections typically represents the connection to the centre settlement (villages have only fewer network connections); thus, it is clear that small settlements closer to the nodes of the settlement network are more active in using these opportunities. The strengthening of local society and local identity is present with great emphasis everywhere when examining international examples. Therefore, it is not valid in the studied countries that this is less important in the vicinity of big cities.

In the study of village renewal strategies, the loss of space for developments based on agriculture and natural resources was noticeable; thus, we examined which factors were emphasized in previous research to determine the types of villages and which development elements became cluster-forming variables. It has become clear that the role of (agricultural) land use is becoming less and less important over time. One hundred fifty years ago, settlements were defined by their status and land use; thus, land use was the basis for grouping in the surveys. In the 20th century, agriculture surveys were first reduced to a "yes/no" question and were then refined into a weightless variable.

Examining the variables (and clusters) of the surveys, it can be stated that the importance of the previously decisive role of land use (initially agricultural and later industrial) in the life and development of villages was taken by the proximity of employment, the role of settlement networks and network priorities. In all this, it can be traced that among the variables of the national surveys analysed in our surveys, more and more indicators have emerged that have analysed the economy and development (or success) of the village.

Overall, we found that the choice of the settlement development strategy is greatly influenced by the situation of the settlement network, the available (internal and external) resources and the internal motivation. Different settlement development strategies can both lead to success, but the role of the settlement network fundamentally influences the opportunities. For example, while the villages of large urban agglomerations and small-town catchment areas can be successful with the proper use of situational energies, a successful development project based on internal resources can stand out from similar settlements. Conversely, settlements have to face unfavourable settlement processes in depopulated areas. This is because the implementation of a successful development strategy in these villages and the realization of favourable development dynamics must always be based on internal resources. That is why in the latter group of settlements many so-called "separate" development strategies and, in the absence of other options, agricultural-based (or other local resource-based) developments are still of strategic importance.

The novelty of our research is that, regardless of the current state of the villages, we grouped the villages based on the goals and priorities of the development strategies. While

some of the previous studies sought to examine the situation of villages and possibly typify them, others examined the success of small settlements by examining a smaller group of settlements. In the present research, we focused on defining strategic priorities depending on the settlement network situation.

## 5. Conclusions

Our article wanted to examine the village development strategies and analyse and evaluate the strategic directions, primarily through the examples of Hungarian villages that participated in the Hungarian Village Renewal Award competition. To understand the objectives of the village development strategies, it was essential to take into account the changes in the Hungarian settlement network and the changed village roles, as well as to become acquainted with the most important goals of the Hungarian and European Village Renewal Award and the competition qualification system. Therefore, we carried out its research by analysing the applications of 50 Hungarian villages, examining how each element of the village renewal strategy relates to the settlement network's role and the distance from the centre settlement.

A significant result of our research was the definition of three groups of settlements according to the position of the villages in the settlement network. After analysing the groups, we found that the villages set different development priorities in terms of the role of the settlement network. Therefore, the villages of each settlement group have a strategy based on different development directions. In the vicinity of large- and medium-sized cities, which are the focal points of the settlement network, development areas for improving the quality of life (e.g., quality building stock, socio-cultural life) dominate, and employment is the most critical area in urban areas, but also strengthening local identity and community building.

Our studies have highlighted the role of agricultural land use in village development strategies. Although agriculture is still an essential aspect of the development of depopulated areas, it is worth considering that the agricultural land is a significant development priority in only a group of villages, serving an increasingly small section of society.

The main goal of our research was to help the villages to survive, renew and develop through the analysis of strategies. To this end, we examined in detail the development measures and strategies of the villages. By arranging the information, we tried to achieve a result that could be used in practice. Based on these, we recommend:

- to provide as much information as possible to the rural municipalities to explore the local conditions more precisely and make a conscious strategy;
- to adapt the support framework of the existing tender resources (with the related indicators) to the villages' groups according to the settlement network's role. Thus, targeted subsidies corresponding to different development priorities would be available to the villages of each settlement group.

**Author Contributions:** Conceptualization, S.B. and Á.S.; Data curation, S.B.; Investigation, S.B.; Methodology, S.B. and Á.S.; Resources, S.B. and Z.S.; Writing—original draft, S.B. and Z.S.; Writing—review & editing, Á.S. All authors have read and agreed to the published version of the manuscript.

**Funding:** This research received no external funding.

**Data Availability Statement:** The applications of the Hungarian and the European Village Renewal Awards are available digital format by the authors.

**Conflicts of Interest:** The authors declare no conflict of interest.

## Appendix A  Motto and Winners of the European Village Renewal Competitions

**Table A1.** Motto and winners of each competition.

| | |
|---|---|
| 1990: International exchange of experiences | Dorfbeuern\|Salzburg\|Austria |
| 1992: Being there is everything | Illschwang\|Bavaria\|Germany |
| 1994: Own initiative is trump | Steinbach an der Steyr\|Upper Austria\|Austria |
| 1996: Comprehensive village renewal | Beckerich\|Luxembourg |
| 1998: Creative-innovative-cooperative | Obermarkersdorf\|Lower Austria\|Austria |
| 2000: There is no past without future | Kirchlinteln\|Lower Saxony\|Germany |
| 2002: Crossing borders | Großes Walsertal\|Vorarlberg\|Austria |
| 2004: Meeting the challenge of uniqueness | Ummendorf\|Saxony-Anhalt\|Germany |
| 2006: Change as opportunity | Koudum\|Netherlands |
| 2008: Win the future through social innovation | Sand in Taufers\|South Tyrol\|Italy |
| 2010: New energy for a strong togetherness | Langenegg\|Vorarlberg\|Austria |
| 2012: On the track to the future | Vals\|Grisons\|Switzerland |
| 2014: Lead a better life | Tihany\|Veszprém\|Hungary |
| 2016: Open mind | Fließ\|Tyrol\|Austria |
| 2018: Th!nk further | Hinterstoder\|Upper Austria\|Austria |
| 2020: Local answers to global challenges | Municipal Alliance Hofheimer Land\|Bavaria\|Germany |

## Notes

[1] Health and social care facilities are available in Bad Liebenstein and are provided by mobile public transport, e.g., E-car, E-carriage, village bus.

[2] Hungarian statistician, geographer, teacher, professor at the University of Pest.

[3] Hungarian statistician, writer of economic statistics and geography, member of the Hungarian Academy of Sciences.

[4] Researcher (regional sciences), university professor. His main field of research is the historical and settlement geography of Hungary.

[5] Researcher (regional sciences), university professor. He specializes in marketing geography.

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
