# Peer review of "Strategic Directions: Evaluation of Village Development Strategies in the Case of Applicants for the Hungarian Village Renewal Award"

_land, doi:10.3390/land11050681_

Round 1

Reviewer 1 Report

I would recommend redesigning the research and focusing on what is written in the title: Strategic directions: Evaluation of village development strategies in the case of applicants for the Hungarian Village Renewal Award.

In this context, the part describing the history “3.2. Declining importance of rural land uses” (409-532) needs to be thoroughly revised, as it is not clear how it relates to “village development strategies in the case of the Hungarian Village Renewal Award“.

Figure 4 would make sense when compared to maps from later periods, Figures 5 and 6 are not clear because they do not explain what types of villages the colors mean. It is not clear what the type changes are and how they relate to current strategies. This is not sufficiently described in the discussion section either.

Reviewer 2 Report

Evaluation of village development strategies are important for many decisions made at regional levels. The authors utilized historical materials and related materials for the Hungarian Village Renewal Award competition to present the European and Hungarian village renewal movement. Although the article is informative, it is more like a history of village development than an academic study. In particular, the research scientific questions are not clear and the research is too diffuse. Here are some detailed suggestions.

1)The content of the article is scattered. it is recommended to use a logic diagram or flowchart to link up the main sections.

2)The section of introduction details The European Village Renewal Award, but it doesn't tie in well with the first part of Changed village roles. It is recommended to add a paragraph summarizing the relationship between two parts.

3)The main conclusions of the article are not very much connected to the research process and can all be derived directly from historical materials.

4)It is recommended to indicate the sources of the data and official materials involved in the study. Some of the materials need to be accompanied by links to relevant web pages.

5)The article lacks a basic introduction to the study area, especially the land use and socio-economic development of the settlement.

6)The discussion section does not provide a good in-depth discussion of the main findings in conjunction with related previous studies.

Reviewer 3 Report

Effective evaluation of village development strategies is important for clarifying village positioning and promoting sustainable village development. Starting from theoretical construction and typical case studies, this paper analyzes the significant impact of village renewal on rural development. The logic is clear, and the conclusion is credible. I suggest a minor revision. The opinions are as follows:

  1. The abstract needs to be further condensed to highlight the scientific significance and value of the study.

2.Part 3.1 outlines the strategic development orientation of rural settlements in other countries, but the overall feeling is more like a pile-up, lacking a condensed summary, where the logical relationships need to be further sorted out, and it is recommended to further refine the similarities and differences between international rural development and the study case area Hungary.

3.The figures of the article are provided in clear versions or redrawn as much as possible, especially the cited Figures 4, 5, and 6.

  1. The article divides the sample settlements into three categories, the scientific validity of which is subject to further examination.
  2. The discussion section suggests further deepening the research content and outlook to enhance the academic value of the study. Can some practical development suggestions be made for villages in different stages of development and different zones of Hungary?

6.The format of some references is not standard and should be revised.

7.The overall logic and science of the article's exposition needs further improvement.
